# 3D Printing of High Viscosity Reinforced Silicone Elastomers

**DOI:** 10.3390/polym13142239

**Published:** 2021-07-08

**Authors:** Nicholas Rodriguez, Samantha Ruelas, Jean-Baptiste Forien, Nikola Dudukovic, Josh DeOtte, Jennifer Rodriguez, Bryan Moran, James P. Lewicki, Eric B. Duoss, James S. Oakdale

**Affiliations:** 1Materials Engineering Division, Lawrence Livermore National Laboratory, 7000 East Avenue, Livermore, CA 94550, USA; nick.rodriguez@utexas.edu (N.R.); dudukovic1@llnl.gov (N.D.); deotte1@llnl.gov (J.D.); moran5@llnl.gov (B.M.); lewicki1@llnl.gov (J.P.L.); duoss1@llnl.gov (E.B.D.); 2Department of Mechanical Engineering, The University of Texas at Austin, 204 E. Dean Keeton Street, Austin, TX 78712, USA; 3Materials Science Division, Lawrence Livermore National Laboratory, 7000 East Avenue, Livermore, CA 94550, USA; ruelas7@llnl.gov (S.R.); forien1@llnl.gov (J.-B.F.); rodriguez96@llnl.gov (J.R.)

**Keywords:** stereolithography, 3D printing, silicone, elastomer, MQ resin, thiol-ene

## Abstract

Recent advances in additive manufacturing, specifically direct ink writing (DIW) and ink-jetting, have enabled the production of elastomeric silicone parts with deterministic control over the structure, shape, and mechanical properties. These new technologies offer rapid prototyping advantages and find applications in various fields, including biomedical devices, prosthetics, metamaterials, and soft robotics. Stereolithography (SLA) is a complementary approach with the ability to print with finer features and potentially higher throughput. However, all high-performance silicone elastomers are composites of polysiloxane networks reinforced with particulate filler, and consequently, silicone resins tend to have high viscosities (gel- or paste-like), which complicates or completely inhibits the layer-by-layer recoating process central to most SLA technologies. Herein, the design and build of a digital light projection SLA printer suitable for handling high-viscosity resins is demonstrated. Further, a series of UV-curable silicone resins with thiol-ene crosslinking and reinforced by a combination of fumed silica and MQ resins are also described. The resulting silicone elastomers are shown to have tunable mechanical properties, with 100–350% elongation and ultimate tensile strength from 1 to 2.5 MPa. Three-dimensional printed features of 0.4 mm were achieved, and complexity is demonstrated by octet-truss lattices that display negative stiffness.

## 1. Introduction

Soft, elastomeric ‘rubbery’ materials are polymers with low elastic moduli and high deformation capability, which make them ideal candidates for various applications, including shock absorption, cushions, seals and gaskets, tires, tubes, and belts. Elastomeric feedstocks are actively being developed for additive manufacturing (AM) [1] and are being applied to soft robotics [2], metamaterials [3], auxetic materials [4], and prosthetics [5]. Despite tremendous progress over the past 15–20 years, only a subset of traditionally available materials is 3D-printable, and AM-feedstock development remains a considerable challenge. In an ideal scenario, the underlying properties of an AM material would be identical to those obtained via traditional manufacturing (subtractive, injection molding, etc.). However, 3D-printed materials tend to display lower mechanical properties, inferior aging, and anisotropic behavior, all of which limit AM for end-use production [6,7]. Feedstock development challenges hinge on maintaining desired properties while adapting the material to conform to the engineering constraints of a particular printer and/or printing process. As examples, this may include endowing a material with UV-reactive functional groups for a stereolithography (SLA) process [8] or formulating with thixotropic additives to facilitate direct ink writing (DIW) [9]. Here, we have approached this problem by co-developing a UV-curable silicone elastomer with the build of an SLA printer specifically designed to handle the viscous nature of reinforced silicone resin.

Silicones are an important class of semi-inorganic polymers valued for their unique combination of desirable properties, including exceptional thermal stability, chemical inertness, and a unique ability to maintain useful properties over a wide range of service temperatures (−50 °C to over 250 °C) [10]. Elastomeric silicone rubbers generally consist of networks of polydimethylsiloxane (PDMS) or poly(methyl/phenyl)siloxane copolymer chains crosslinked together using various chemical strategies, including condensation, free-radical coupling, or Pt-catalyzed hydrosilylation [11]. Nearly all high-performance commercial silicone rubbers are composites consisting of siloxane networks with embedded reinforcing particulate filler [12]. Commonly employed fillers include silica (fumed, colloidal), carbon black, and diatomaceous earth. The incorporation of sub-micron-sized particulate filler significantly improves mechanical properties and increases abrasion resistance in rubbery networks [13]. Without filler, silicone vulcanites are extremely soft and weak and have limited use in mechanical applications. 

DIW and ink-jetting are the preferred methods for 3D-printing silicone parts and have recently been commercialized [14,15]. However, there are many advantages in developing a lithographic-based approach, including unparalleled resolution and throughput, as evidenced by SLA’s role as an enabling technology at the heart of the multi-billion dollar per year dental aligner industry [16]. Moreover, in comparison to DIW technologies, SLA enables the design and build of more complex structures, including architectures with unsupported overhangs. Recently, several groups have reported on the SLA of silicones [17,18,19,20,21,22,23,24,25,26,27,28] and/or related polysiloxane-containing materials [2], using predominately acrylate [23,24,25,27] and/or thiol-ene [17,18,19,20,21,22,26] crosslinking as workhorse chemistries. Notably, however, many reported SLA silicone formulations have lacked the addition of reinforcing filler. Presumably, this is because the addition of filler sharply increases the viscosity of silicone formulations, which challenges the development of traditional SLA-based processing [18].

SLA photoresists are generally designed to have low starting viscosities to facilitate layer-by-layer recoating, i.e., uncured resin wets the surface of cured material at each step of the printing process [1]. Materials with higher surface tensions or viscosities simply do not level out to form thin layers. This issue has been addressed in part by using bottom-up approaches, in which thin layers of resin are formed by lowering a build plate into a resin bath. Transparent baths allow for thin layers of resin to be illuminated from below. While this approach has proven successful [29], it does introduce the additional problem of needing to separate each printed layer from the bath floor. If unmitigated, the forces required to separate a printed layer from the bath floor can damage small features or even tear the printed structure from its build substrate. Multiple solutions to this problem have been explored with varying degrees of success, including tilting the bath, translating the bath laterally, and using a flexible release film or oxygen-permeable membrane as the bath floor, such as the CLIP process [30,31,32,33]. 

This work presents a custom-built bottom-up digital light projection SLA printer designed specifically to process paste-like (up to 100 kPa·s) silicone resins. The paste-SLA printer was then utilized to develop a series of tunable UV-curable silicone elastomers based on thiol-ene crosslinking and reinforced with a combination of MQ resins and fumed silica. 

## 2. Materials and Methods

### 2.1. SLA-Paste Printing

The paste-SLA printer utilizes a DLP6500FLQ 0.65 1080p DMD (TI, Dallas, TX) to pattern a 14.5 × 8.16 mm^2^ light field of 405 nm light with a pixel size of 7.56 µm. This image is magnified 2× before passing through a transparent vat made of 0.127 mm FEP Teflon. A mechanical aperture in the optical path limits the intensity of light that travels to the build plane, and a decreasing aperture diameter can increase the resolution of the patterned light. Appendix A contains experimental measurements relating aperture size to light intensity at the build plane. FEP Teflon was chosen to provide a low-surface energy interface. A LTS150 translation stage and CR1-Z7 rotational stage (Thorlabs, Newton, NJ, USA) are controlled through LabVIEW (NI, Austin, TX, USA) and translate the build substrate vertically and rotate the resin bath for recoating, respectively. The resin bath is modeled after the FlexVat [34] and allows for varying the tension of the FEP Teflon film. At lower tension, the film can deform to larger peel angles.

Printed structures with 50 µm layers were fabricated using patterned 405 nm light with an intensity of 34 mW/cm^2^ for 7 s per layer. After printing, parts were cleaned of residual uncured resin by first compressing the structures to extrude the bulk of the resin out of the structures’ inner pores. Parts were then placed in a mixture of Dawn dish soap detergent and water and agitated by hand until the part was deemed visually clean. Parts without pores, such as tensile specimens, were wiped clean of excess resin using kim-wipes. 

### 2.2. Materials

All materials were used as received unless otherwise noted. Vinyl terminated polydimethylsiloxanes (VTS-1, -2, -3), (mercaptopropyl)methylsiloxane-dimethylsiloxane copolymers (MFS-1, -2), and amphorous hexamethyldisilazane treated fumed silica (surface area = 200 m^2^/g) were obtained from Gelest, Morrisville, PA, USA. Isopropylthioxanthone (ITX), 4-methoxyphenyl (MEHQ), 2-ethylhexyl 4-(dimethylamino)benzoate (EHDA), 2-(2H-benzotriazol-2-yl)-4,6-di-tert-pentylphenol (BTA), tetraorthosilicate, hexamethyldisiloxane, sodium bicarbonate, sodium sulfate, and sulfuric acid were obtained from Sigma Aldrich, St. Louis, MO, USA. Tetrahydrofuran (THF), ethanol, toluene, dichloromethane (DCM), and isopropanol were obtained from VWR, Radnor, PA, USA as ACS grade. All deuterated solvents for NMR analysis (chloroform, toluene, etc.) were obtained from Cambridge Isotopes, Tewksbury, MA, USA. FEP film for the resin bath was obtained from American Durafilm, Holliston, MA, USA.

MQ resin Synthesis: MQ resin was synthesized according to a procedure described by Flagg et al. [35]: A 500 mL round bottom flask equipped with a stirring bar and reflux condenser was charged with 90 mL of ethanol, 50 mL water, and 50 µL sulfuric acid. The reaction vessel was then warmed to 50 °C in an oil bath. Tetraorthosilicate (60 g, 0.29 mol) was added, and the reaction mixture was allowed to stir at 50 °C for 2 h. A combined mixture of hexamethyldisiloxane (21.9 g, 0.13 mol) and 1,3-divinyltetramethyldisiloxane (2.9 g, 0.016 mol) was then added, followed immediately thereafter by 2 mL of sulfuric acid. The resulting mixture was then gradually warmed to 80 °C while stirring over the course of 2 h. The reaction vessel was removed from the oil bath, allowed to cool to room temperature, and the contents were transferred to a separatory funnel. The lower layer (organics) was collected and washed with saturated sodium bicarbonate and then dried over magnesium sulfate. Rotary evaporation, followed by 70 °C vacuum oven drying, yielded a white powder. The M-to-Q ratio of the powder was determined to be approximately 0.8 via NMR analysis following a procedure described by Wu et al. [36]. The vinyl concentration of the resin was also determined by NMR analysis to be approximately 0.5 mmol/g. 

Resin Formulation: VTS-1,-2,-3 and MFT-1,-2 were mixed at a stoichiometric ratio of 1:1 vinyl-to-mercapto function groups as prescribed in Table 1. Next, 0.6 wt% EHDA was added to the formulation, followed by a solution of 0.3 wt% ITX, 0.1 wt% MEHQ, and 0.24 wt% BTA dissolved in 0.5 mL THF. The subsequent mixture was then subjected to planetary centrifugal mixing using a Thinky. TEFS formulations: Fumed silica was added batch-wise a few grams at a time with intermediate Thinky mixing after each addition. TEMQ formulations: MQ resin was dissolved in a minimal amount of DCM (approximately 5 mL DCM per gram of MQ resin) and mixed by hand with the desired amount of VTS. The resulting mixture was then placed under a stream of nitrogen for 12 h or until the DCM was removed as determined gravimetrically. MFS, EDHA, ITX, MEHQ, BTA, and fumed silica were then added as described above. Resin formulations containing MEHQ inhibitor exhibited shelf-lives of 12 months or more.

### 2.3. Instrumentation

Tensile testing was conducted on an Instron Mechanical Testing System according to ASTM D638. A series of 3 mm thick sheets of ‘bulk’ test specimens were cured with a Loctite LED flood array with 405 nm wavelength at 250 mW/cm^2^ and cut with a type IV dumbbell specimen tensile die. Bulk specimens were mounted with pneumatic grips and measured against a 100 N load cell at a crosshead speed rate of 10 mm/min. Strain was recorded as the displacement between the crossheads, and Young’s modulus was calculated from the resulting stress–strain curve over 0–2%. SLA-printed tensile specimens were 20% the size of the ASTM geometry and were printed with their shortest axis oriented in the Z direction. A pull rate of 15 mm/min was used to test these specimens to failure. Strain was recorded by marking the gauge area with permanent marker recording displacement with an in-line camera. 

Compression testing of the printed lattices was also conducted on the Instron MTS, and a compression rate of 5 mm/min was used to compress the structures to increasing strains from 33% to 87%.

Shore A hardness experiments were carried out with a PTC Instruments Durometer Model 408. NMR spectroscopy was conducted on a 600 MHz Bruker, Billerica, MA, USA spectrometer. All features were measured using a Zeiss, Oberkochen, Germany, Stereo Discovery V8 stereomicroscope.

Rheology experiments were performed on a TA Instruments, New Castle, DE, USA, AR2000ex rotational rheometer using a 20 mm parallel plate setup. All measurements were carried out using a sample thickness (gap) of 1 mm. Oscillatory measurements of the elastic (G′) and viscous (G″) shear moduli were performed as frequency sweeps at a low constant strain of 0.05%. The values of G′ were reported at a frequency of 1 Hz. To test the thermal stability of the resin, oscillatory experiments were carried out at temperatures 25–150 °C. The temperature was controlled using a Peltier plate, and the resin was allowed to equilibrate at the set temperature for 2 h before the measurement. Shear viscosity was measured as a flow ramp by varying the shear rate from 10 to 2 to 102 s^−1^.

Differential scanning calorimetry (DSC) was performed on the cured samples to determine the transition temperature of the silicone materials using a Cryo-Discovery series DSC (TA Instruments). The samples were subjected to a heat, cool, heat cycle that ramped from −140 to 50 °C at 10 °C min^−1^, held at 50 °C for 1 min, ramped down to −140 °C at 10 °C min^−1^, and heated again at a rate of 10 °C min^−1^ up to 50 °C.

X-ray tomography imaging was performed at beamline 8.3.2. of the advanced light source, Berkeley, CA, USA [37], tuned to illuminate the gyroid sample with a 22 keV X-ray beam. Projections were collected through a 10× lens onto a 2560 × 2160 pixel array PCO edge 4.2 detector, resulting in a resolution of 6.16 µm/pixel. A total of 1313 radiographies were collected around a 180° rotation range with an exposure time of 1 s. A series of 5 vertical scans were required to image the entire height of the sample. Normalization of images was carried out with standard flat-field correction, followed by filtered back projection and downscaling of reconstructed volume with a factor of 3. Using Fiji image processing [38] package 1.53c [39], a 2D median filter (2 pixels radius) was applied for noise reduction and image segmentation was applied using Otsu’s method to differentiate the sample from the surrounding air. The extracted sample was converted to STL file format using 3D slicer 4.8.1 [40] and registered and compared to a prescribed file using CloudCompare 2.12 [41].

## 3. Results

### 3.1. DLP-SLA Paste Printer Design and Build

An SLA paste-printer was designed with a bottom-up approach using a reconfigurable digital mask to pattern a uniform field of light from a 405 nm LED array, see Figure 1. The resin is illuminated from beneath through a transparent vat floor with a maximum intensity of 34 mW/cm^2^ over an area of approximately 29 × 16 mm^2^. 

In this technique, the build plate and printed structure descend from above the vat to trap a thin layer of resin in between the structure’s previously cured layer and the vat floor. Thin layers (100 µm or less) can be achieved with high-viscosity resins and even pastes, and layer thickness is only limited by the degree of coplanarity between the build substrate and the vat floor. Figure 1a shows the overall printing process: (1) the build plate is lowered into a thin layer of resin, (2) a pattern is projected from below, and (3) the structure is then lifted out of the vat and the vat is rotated 30° to present a new area of uncured resin [29]. 

Finally, thin layers of uncured resin were crucial for mitigating part deformation, especially for higher viscosity resins, displacement of which can exert significant forces on the structure as it is lowered into the resin bath. Thin layers are achieved by incorporating a stationary wiper blade with a gap thickness of approximately 500 µm, see Figure 1c. 

### 3.2. Formulation of UV-Curable Silicone Elastomers 

The silicone formulations in this study were developed using vinyl terminated-polydimethylsiloxane (VTS) and polymercaptosiloxane-*co*-dimethylsiloxane (MFS) monomers as crosslinkers, see Scheme 1. UV-initiated thiol-ene crosslinking of silicones was first described in the 1970s [42] and has since seen renewed interest for 3D printing [17,18,19,20,21,22,26]. A wide variety of VTS monomers are commercially available, and although the selection of mercapto-containing siloxanes is more limited, several synthetic approaches for their preparation have been described in recent years [23,43]. 

Overall, the formulation of thiol-ene silicones was challenged by a lack of miscibility of MFS in VTS and general poor solubility of photo-initiators in PDMS (see Appendix A). Mercapto-siloxane copolymers (MFS) containing greater than ca. 10 mol% mercaptopropyl methylsiloxane relative to dimethylsiloxane were found to have only limited solubility in vinyl-terminated polydimethylsiloxane. Initial formulations were carried out with commercially available MFS-1, which has manufacturer specified 4–6 mol% thiol functional groups, determined to be 0.485 mmol/g via ^1^H NMR analysis. The molar ratio of thiol-to-vinyl functional groups was held at a 1:1 for all formulations, see Table 1. The photo-initiator package consisted of a type II radical initiator (ITX + EDHA), a photo-absorber (BTA) and a radical inhibitor (MEHQ). 

Varying amounts (phr) of reinforcing hexamethyldisilazane treated fumed silica with a vendor specified surface area of 200 m^2^/g were used for reinforcement. Silicone formulations containing fumed silica are designated as TEFS. The viscosity of a VTS-2 + MFS-1 formulation rose from 4 Pa·s (TEFS-1, Table 1) to over 100,000 Pa·s (TEFS-4) with a compounding of 33 phr (see Appendix A). In addition to filler, the mechanical properties of siloxane networks can be tuned by controlling the extent of crosslinks and the distribution of chain lengths [11]. However, the solubility of MFS-2 (1.64 mmol/g) was found to be less than 3 wt%, severely hampering our ability to tune the degree of crosslinking. This issue was addressed in part by adding vinyl-functionalized MQ resin, which we found enhanced miscibility of MFS-2 in VTS-2 and VTS-3. MQ resins are nm-sized inorganic/organic hybrid molecules consisting of a SiO_4/2_ network (‘Q’-unit) decorated with trialklylsilyl functional groups (‘M’-unit) [36,44,45]. These materials have a particle-like morphology yet display excellent solubility in most organic solvents, including PDMS. MQ resin has found use as a reinforcing agent in optically transparent silicone formulations [46]. The MQ resins utilized in this work were synthesized from tetra orthosilicate and had an M-to-Q ratio of 0.8 and vinyl concentration of approx. 0.5 mmol/g. Silicone formulations containing MQ resin are designated as TEMQ.

### 3.3. Mechanical Properties and Testing

Miniaturized ASTM D638 type IV dogbone specimens were printed at 80% power (405 nm, 27 mW/cm^2^, 7 s per layer) in the XY plane with 50 µm steps (smallest dimension in the Z-axis) within the maximum available build window of our paste-SLA printer. This corresponds to a 3D-printed specimen with dimensions that are approximately 20% of the ASTM specifications. Elongation was measured by tracking the displacement of fiducial markers added to the gage section of the specimen (Figure 2c). Both engineering stress and strain are reported for all samples in Table 2, and we note that we are likely over-estimating the true tensile strength at break due to difficulties associated with gripping these small samples. A series of corresponding ‘bulk’ specimens were prepared to validate the tensile results obtained for printed TEMQ1–4. Bulk samples of 3 mm thickness were cured under a 405 nm 250 mW/cm^2^ flood lamp for 10 min from which full-size type IV dogbones could be die-cut. The results of the printed and bulk specimens were statistically similar, giving confidence to the results obtained from smaller 3D-printed samples (see results in Appendix A). The addition of fumed silica from 0 to 33 phr resulted in stronger, tougher materials; see TEFS-1 through 4 and Table 2 (for corresponding stress–strain curves, see Appendix A). Finally, the thermal properties of thiol-ene cured TEFS1–4 were evaluated under DCS and found to exhibit stereotypical PDMS crystallization onset at −75 °C and glass transition at ca. −115 °C (see Appendix A). 

The similarities between the cured and ‘bulk’ specimens were also surprising, given the difference in light intensities, 27 vs. 250 mW/cm^2^, respectively. We further explored post-cure strategies as a means to improve the mechanical properties. Prolonged post-UV exposure (>hours) did not significantly improve the tensile properties of 3D-printed specimens, although the materials did noticeably yellow. Post-print thermal exposure was even more complicated. Short durations at 180 °C (<2 h) also did not improve the mechanical properties but did cause yellow/browning. Prolonged thermal exposure (>24 h) or thermal cycling negatively impacted material properties and notably induced embrittlement and friability even in samples with high fumed silica loading (i.e., TEFS-4). The mechanism leading to embrittlement is unknown, and we are currently evaluating methods to track the degree of cure in an effort to address this issue. 

The effect of the VTS chain length was evaluated for three samples, TEFS-4, 5, and 6, each with 33 phr fumed silica, see Figure 2a. There is a noticeable increase in toughness moving from VTS-1 (0.61 mmol/g vinyl) in TEFS-5 to VTS-2 (0.12 mmol/g) in TEFS-4. The mechanical properties of the TEFS-5 formulation with 33 phr fumed silica are similar to the properties obtained from TEFS-1 with 0 phr fumed silica, Table 2. This result highlights that the optimization of silicone formulations is a function of both the underlying network structure and its interaction with the reinforcing filler. 

The incorporation of MQ resins was envisioned as a means to reduce the viscosity of the resin while still maintain high levels of reinforcement. The shear modulus (G′) of the resins was measured as a function of filler; see Table 2 or Appendix A for a graphical representation. Increasing concentrations of silica from 0 to 33 phr result in an exponential rise in G′ (TEFS-1 to 4). We were able to achieve similar G′ values for TEMQ-2 and -4 compared to TEFS-1 with higher combined MQ resin plus fumed silica filler concentrations, 50 and 69 phr, respectively, vs. 33 phr. 

As noted in Section 3.2, the addition of MQ resin also enhanced the miscibility of MFS-2. The addition of vinyl functionalized MQ resin (ca. 0.5 mmol/g) and MFS-2 (1.64 mmol/g thiol) improved tunability and enabled formulations with varying hardness, stiffness, and strength, see Figure 2b. Increasing the concentration of MQ resin generally resulted in stiffer, harder parts that failed at higher ultimate tensile strengths (σ_max_). The greatest σ_max_ was achieved with TEMQ-4 at around 2.5 MPa. These results are likely directly correlated with increased crosslinking. Notably, formulations with only MQ resin and performed poorly, see TEMQ-3. Although the exact mechanism of MQ resin reinforcement is unknown, Kishi et al. have found evidence of MQ resin clustering/domain formation in PDMS/MQ resin blends [47]. We hypothesize that MQ resin clustering in effect results in the formation of larger agglomerates that provide network reinforcement. In this case, fumed silica may be needed to nucleate MQ resin domain growth. TEMQ-1 with a combined 33 phr MQ resin and fumed silica significantly out-performed TEFS-4 containing only 33 phr fumed silica. TEMQ-1 exhibited nearly twice σ_max_ with an order of magnitude lower G′. 

Finally, cyclic tensile experiments were carried out on TEFS-4 at a pull rate of 2 mm/s to 60% of elongation at break (Figure 2d). The stress-strain curve exhibits a drop in the elastic modulus following the first cycle, characteristic of the Mullins effect (bold green dotted line). The specimen was then observed to continue to relax, albeit at a slower rate, over the next 40 cycles, before ultimately failing due to the formation of a crack in the gage section. 

### 3.4. Optimization of SLA-Paste Printing

Refinement of the printing process was carried out on TEFS-2 for ease of handling under a variety of conditions, including exposures time (5–10 s), light intensity (7–34 mW/cm^2^), and photo-initiator concentration. Features and voids were evaluated with the test mask shown in Figure 3a, which consists of pairs of 500 µm lines arranged with a varying pitch from 0.25 to 3 mm. All resins were formulated with a type II photo-initiator consisting of 0.3 wt% ITX and 0.6 wt% EHDA. A MEHQ inhibitor was added to both to enhance the shelf-life of the resin and for tuning lateral resolution. Increased amounts of MEHQ (0.05 vs. 0.1 wt%) generally resulted in sharper voids (Figure 3a,b) and finer features (Figure 3c). The only voids not over-polymerized (i.e., the measured area was less than programmed area) were fabrication with 0.10 wt% MEHQ and lower light intensities 20 mW/cm^2^. Voids of 0.25 × 0.25 mm^2^ or less were not able to be resolved. Line widths generally increased as pitch decreased, except again in the case of higher inhibitor and lower intensities (solid squares, Figure 3c). Photo-absorber (BTA) was also tuned to decrease layer thickness, as seen in Appendix A. A layer thicknesses of approximately 0.3 mm could be obtained at 0.5 wt% BTA loading. 

A 1 × 1 × 1 cm^3^ gyroid sample made with TEFS-3 with 24 mW/cm^2^ (Figure 4a) was imaged by X-ray tomography to investigate defects introduced in the object during the printing process. No porosity was found in the walls of the gyroid structure, suggesting proper adhesion of the successive printed layers. In addition, slices of the reconstructed gyroid volume show unobstructed channels (Figure 4b). Further analysis was performed to compare the prescribed geometry to the final printed object. Triangulated surfaces of the input geometry and reconstructed volume were aligned in 3D (Figure 4c), and distance differences of the triangles from both volumes were measured. Results indicate that the printed object is, on average, slighty bigger than originally intended (0.266 mm) with local variation. This result is consistent with the tendency of voids towards over polymerization. 

### 3.5. Negative Stiffness Behavior in Printed Lattices

A series of compression tests were conducted on lattice structures with an octet truss unit cell to observe their mechanical performance and negative stiffness behavior. Lattices were printed from TEMQ-3 and subjected to increasing compressive strains from 33% to 87% before failure occurred at the nodes, and the lattice did not recover. Figure 5 displays stress vs. strain data along with images of the lattice’s behavior during compression up to 67% strain. There is a period of negative stiffness between 16% and 20% strain when one layer of struts buckles and the unit cells collapse, and an additional period of negative stiffness that occurs at 35% strain when the second-row collapses. After these two regions of negative stiffness, the stress has a positive slope as it approaches full densification, with failure occurring at 87% strain and 2.4 MPa (Appendix A). The negative stiffness regions at which struts start to buckle are tunable by adjusting strut thickness, so structures such as these could be designed to isolate shocks based on expected force amplitude. The two regions of negative stiffness seen in this structure, for instance, could be adjusted so that one region occurs at relatively low forces and another at higher forces to accommodate a range of possible impacts. 

## 4. Discussion

The objective of this work was to develop an SLA methodology to produce end-use silicone parts of arbitrary complexity. We developed a paste-printer design specifically to handle non-Newtonian filler-reinforced silicone resins. The results of this work highlight several challenges that are actively informing and guiding our subsequent efforts in this area. 

First, although the thiol-ene click chemistry is an attractive crosslinking strategy for silicones, two issues have presented significant hurdles for future development. As noted in Section 3.3, we observed peculiar and rather poor aging behavior upon exposure to elevated temperatures. Silicones are renowned for having a wide service-temperature window, including above 200 °C. The observed embrittlement behavior is currently under investigation. Further, although we were able to demonstrate tunability (Figure 2), we thus far have been unsuccessful in reaching tensile properties found in commercial-grade silicones cured through alternative methods (σ_max_ > 5 MPa, ε_max_ > 400%). We believe that part of the challenge associated with developing thiol-ene silicones is related to the solubility, or lack thereof, of the mercapto functionalized monomer (MFS) in vinyl terminated PDMS. MQ resin provided a partial solution as this material was found to improve MFS solubility. During the writing of this manuscript, we became aware of at least two other groups that took a similar MQ resin approach for the development of SLA silicones. In 2007, Shin-Etsu chemical Co. patented an MQ resin reinforced resin [48], and in 2019, Zhao et al. reported a formulation with an MT resin [20]. Neither group reported significantly improved mechanical properties to those demonstrated in this work. 

Significant challenges also remain in the recoating/plunging and lift-off phase of the printing process (i.e., Steps 1 and 3 in Figure 1a). For some of the highest-viscosity silicone resins, the stiffness of the uncured material is similar, if not greater, than the stiffness of the printed structure, particularly for lattices with finer features (Figure 5). In the current paste-SLA design, the printed structure needs to displace excess resin in the vat during the plunging phase. During this process, the printed structure can be compressed or deformed laterally, and both scenarios result in errors and defects during illumination. Separation forces during lift-off continue to present a significant hurdle. Despite our efforts to use a low surface energy release film as the vat floor, some structures could have small features torn during the separation process, and in other cases, the entire structure could be torn off the build substrate. An example of this type of damage can be seen in the frayed edges on the right-hand side of the octet truss lattice (Figure 5). The currently available techniques for reducing this separation force, such as tilting the build plate or only loosely tensioning the release film, are limited to creating relatively small peel angles between the release film and the cured layer. With this in mind, we are exploring solutions that can create larger peel angles, and we are exploring alternative strategies such as acoustic vibration to reduce the separation force.

## 5. Conclusions

In summary, we have developed a series of UV-curable thiol-ene silicones reinforced by a combination of fumed silica and MQ resin. The resulting silicone elastomers are demonstrated to have tunable mechanical properties, with ultimate tensile strength and elongation up to 2.5 MPa and 350%, respectively. A custom-built SLA printer was co-developed and built specifically to process the paste-like silicone resins. Features of 0.5 mm were reliably achieved, and complexity was demonstrated through the fabrication of gyroid and octet-truss lattices. We are in the process of scaling this technology through the design and build of a larger printer with the targeted goal of producing parts as large as 15 × 15 × 15 inch^3^. The next-generation printer and corresponding materials will be used to meet the needs of internal programs for the production of complex gasketing and soft robotic components.

## Data Availability

The data presented in this study are available in Appendix A. Raw data files are available on request from the corresponding author.

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
