# Peer review of "3D Printing of High Viscosity Reinforced Silicone Elastomers"

_polymers, 2021, doi:10.3390/polym13142239_

Round 1

Reviewer 1 Report

The manuscript: 3D printing of high viscosity reinforced silicone elastomers is really interesting, not just for the results but also for the authors have explained every detail.

I do not have many requests about this document.

The dashed line in Figure 2d should be better in green color, similar to Figure 2a for the TEFS-4.

The manuscript is complete. However, the quality of Figures is really poor. I consider the authors need to consider the quality of the entire Figures of this manuscript, This should help to improve definitively their work.

Author Response

We would like to thank Reviewer 1 for their kind words and for taking the time to review our manuscript. We very much appreciate their suggestion to the change the color scheme in Figure 2d to match 2a and we have implemented that change. And finally, we are very thankful that Reviewer 1 brought the issue of figure quality to our attention. We believe we have corrected this problem and have resubmitted a draft with higher resolution figures.

Reviewer 2 Report

this manuscript could be published after revision 1. what is the stability of thiol and ene after mixture? because the thiol and ene will react without UV irradiation. 2. what about heat the washed sample (final parts)? let the thiol and ene have thermal reaction. 3. what is the effect of particle size for the UV polymerization? because of the light scatting.

Author Response

We would like to thank Reviewer 2 for spending time to review our manuscript and they have asked very relevant questions, hopefully our answers below are satisfactory.

Q1. What is the stability of thiol and ene after mixture? because the thiol and ene will react without UV irradiation.

Great comment and question. We have found that our resins are stable for at least a year when stored in the dark and at room temperature.  This is true *if* part of the formulation includes inhibitor. We initially incorporated 4-methoxyquinone (MEHQ) into our material primarily as a means to improve spatial resolution (figure 3). Formulations without MEHQ tended to gel within weeks to a few months, even when stored at room temperature in the dark. We added the following sentence to the manuscript at the end of section 2.2 – “Resin formulations containing MEHQ inhibitor exhibited shelf-lives of 12 months or more.”

Q2. What about heat the washed sample (final parts)? let the thiol and ene have thermal reaction.

Great question again. We had high hopes that we could thermally post-cure our samples to the improve the mechanical properties post-print. However, results of thermal treatment were a bit complicated and surprising in that thermal exposure (>150 oC) was deleterious in some instances. We attempted to address this issue in the manuscript, on page 8/9 we wrote:

“Post-print thermal exposure was even more complicated. Short durations at 180 °C (<2 h) also did not improve the mechanical properties but did cause yellow/browning. Prolonged thermal exposure (>24 h) or thermal cycling negatively impacted material properties and notably induced embrittlement and friability even in samples with high fumed silica loading (i.e. TEFS-4). The mechanism leading to embrittlement is unknown and we are currently evaluating methods to track the degree of cure in effort to address this issue.”   

In the discussion section (page 12/15) we wrote:

“Silicones are renowned for having a wide service-temperature window, including above 200 °C. The observed embrittlement behavior is currently under investigation.”

We still do not understand the observed embrittlement behavior at elevated temperatures. One hypothesis that we have is that potentially residual thiols may attack the siloxane backbone at elevated temps resulting in chain-cleavage. We are still working on quantification of residual thiols in cured material.

Q3. what is the effect of particle size for the UV polymerization? because of the light scatting.

Reviewer 2 is very much correct; particles can be expected to scatter light which can decrease effective dose or result in lower resolution in an SLA or DLP process. We are aware of such studies related to SLA of ceramics (for instance, here is work from 20 years ago: Sensors and Actuators A 1010 (2002) 364-370). Its likely that we will have less amount of scattering in our formulations that contain MQ-resin (TEMQ vs. TEFS) because the MQ-resin dissolves into polydimethylsiloxane homogeneously. However, we have not explicitly carried out scattering studies and therefore have not addressed this phenomenon in our manuscript.